# Proof of Concept for the Detection with Custom Printed Electrodes of Enterobactin as a Marker of *Escherichia coli*

**DOI:** 10.3390/ijms23179884

**Published:** 2022-08-31

**Authors:** Alexandra Canciu, Andreea Cernat, Mihaela Tertis, Silvia Botarca, Madalina Adriana Bordea, Joseph Wang, Cecilia Cristea

**Affiliations:** 1Analytical Chemistry Department, Faculty of Pharmacy, Iuliu Haţieganu University of Medicine and Pharmacy, 4 Louis Pasteur Str., 400349 Cluj-Napoca, Romania; 2Microbiology Department, Faculty of Medicine, Iuliu Haţieganu University of Medicine and Pharmacy, 4 Louis Pasteur Str., 400349 Cluj-Napoca, Romania; 3Department of Nanoengineering, University of California, La Jolla, San Diego, CA 92093, USA

**Keywords:** *Escherichia coli*, enterobactin, analytical methods, electrochemical, wearable sensors, printed electrodes, culture media

## Abstract

The rapid and decentralized detection of bacteria from biomedical, environmental, and food samples has the capacity to improve the conventional protocols and to change a predictable outcome. Identifying new markers and analysis methods represents an attractive strategy for the indirect but simpler and safer detection of pathogens that could replace existing methods. Enterobactin (Ent), a siderophore produced by *Escherichia coli* or other Gram-negative bacteria, was studied on different electrode materials to reveal its electrochemical fingerprint—very useful information towards the detection of the bacteria based on this analyte. The molecule was successfully identified in culture media samples and a future goal is the development of a rapid antibiogram. The presence of Ent was also assessed in wastewater and treated water samples collected from the municipal sewage treatment plant, groundwater, and tap water. Moreover, a custom configuration printed on a medical glove was employed to detect the target in the presence of another bacterial marker, namely pyocyanin (PyoC), that being a metabolite specific of another pathogen bacterium, namely *Pseudomonas aeruginosa*. Such new mobile and wearable platforms offer considerable promise for rapid low-cost on-site screening of bacterial contamination.

## 1. Introduction

Despite the fact that the COVID-19 pandemic has taken a toll on the biomedical world, the detection of bacteria did not pass in a secondary plan, and it remained as a top research topic. From bacterial pneumonia and healthcare-associated infections—often linked to long hospitalization—to water and food contamination, this issue still raises major concern in biomedical and environmental studies, including for patients suffering of COVID-19. The matter is complex and does not require only the rapid identification, but also the selection of the most appropriate antibiotic to avoid inaccurate medication with all the included consequences—namely side effects, diminution of the individual immunity status, and maybe more dangerous, with severe future implications, the selection of resistant bacterial strains.

*Escherichia coli* (*E. coli*) is the most common cause of urinary tract infections, traveler’s diarrhea, neonatal meningitidis, and sepsis produced by Gram-negative bacteria. Pathogenic *E. coli* strains are classified into several pathotypes function of their virulence factors. There are six pathotypes producing diarrhea: Shiga toxin-producing *E. coli* or enterohemorrhagic *E. coli*, enterotoxigenic *E. coli*, enteropathogenic *E. coli*, enteroaggregative *E. coli*, enteroinvasive *E. coli*, and diffusely adherent *E. coli*. The first two types produce a powerful neurotoxin and they are most associated with foodborne outbreaks. Strain O104:H4 (in Europe) and strain O157:H7 (in North America) more frequently become enterohemorrhagic *E. coli* and are capable of initiating outbreaks. They cause severe disease, including dysenteric-like syndrome (bloody-mucous diarrhea) and hemolytic uremic syndrome, a type of kidney failure with thrombocytopenia and convulsions, particularly present in children 0–5 years [1,2,3,4,5]. The emergence of multidrug-resistant strains represents a serious threat and involves an increased risk of complications and death [6,7]. *E. coli* strains mainly reside in the intestinal tract of animal and human hosts and, afterwards, can contaminate water bodies through wastewater effluent and its presence is an indicator of recent fecal pollution. It is important to mention that this bacterial strain has the capacity to survive and grow in environmental conditions for prolonged periods, a concerning issue if it carries antibiotic resistance genes [8].

The U.S. Environmental Protection Agency (EPA) standards and regulations require presence of less than 100 CFU/100 mL, in this case the rapid detection of bacterial strains is a priority to improve the safety profile of the water [9]

According to the Directive (EU) 2020/2184 of the European Parliament and of the Council on the quality of water for human consumption (revised Drinking Water Directive), Annex III, the most important microbiological parameters that should be determined are: *E. coli* and coliform bacteria (EN ISO 9308-1 or EN ISO 9308-2).

ISO 9308-1:2014 specifies a method for the enumeration of *E. coli* and coliform bacteria based on membrane filtration after culture on a chromogenic coliform agar medium, while ISO 9308-2:2012 is based on the growth of target organisms in a liquid medium and calculation of the “Most Probable Number” (MPN) of organisms by reference to MPN tables.

The revised Drinking Water Directive lists for *E. coli* a concentration of 0 CFU/100 mL as minimum requirement for quality of drinking water [10,11].

The golden standard for the detection of bacteria is represented by the well-known culture plates procedures that have a timeframe of approximatively two days into delivering information about the bacterial strain and indicating a specific antibiotic in order to be able to start the right treatment for each patient. This procedure is essential to ensure personalized therapy and to avoid the selection of bacteria with high resistance to antibiotics. The major disadvantage is represented by the long timeframe and the fact that broad spectrum antibiotics are prescribed and could harm the patient and generate further complications. Thus, new methods are necessary to decrease the detection time and to prevent the rise of bacterial resistance [12,13,14,15].

Siderophores are secondary metabolites and have a high affinity for ferric ions. From the almost 500 types, 270 have been characterized and—depending on their structures responsible for iron coordination—they were divided into four main groups, namely: catecholate, phenolate, hydroxamate, and carboxylate. Some mixed structures were also reported in literature [16]. The chemical structure of siderophores and their synthesis, regulation, implication in signal pathways, and pathogenicity of the bacterial strains were thoroughly studied [17,18]. The aforementioned properties, along with the fact that these molecules are electrochemically active, were exploited in the elaboration of various electrochemical sensors for the rapid and direct decentralized detection of the *P. aeruginosa* siderophore, pyoverdine [19,20,21]. An important advantage of this kind of sensing device is represented by the simplicity and the short time lapse for obtaining a result. The challenge is to identify the electrochemical fingerprint of the target and to discriminate the signal among others from complex matrices—such as culture media, water, food samples, and biological samples—without being significantly influenced by the complexity of the sample (low matrix influence). It will represent a turning point for personalized medicine that will force a complete reassessment of antibiotherapy. Recently, the detection of *E. coli* was reported by high affinity sensors—such as immuno/aptasensors—that have a high specificity, but involve a more complicated protocol [22,23,24,25].

While even a qualitative test has the capacity to change the outcome of rapid diagnosis, the final goal is to identify and quantify the bacterial load and link it to the stage of the disease. A library with this information could reshape these strategies and simplify treatment protocols, reduce the economic burden related to medical protocols, and achieve personalized therapy. The detection from culture media after a few hours of incubation will speed up the protocol and will be further integrated in a rapid antibiogram.

The development of wearable sensors has a major impact on conventional protocols and delivers rapid results for on-site analysis, bypassing complicated protocols, and it offers a large spectrum of applications. In-lab printed electrodes have the advantage to use specifically customized inks and to increase the sensitivity and specificity of the detection. Overall, this approach brings the development of electrochemical sensors one step closer to integrated sensors that are able to be used in portable devices [21,26]. The use of medical gloves as a support for the printed electrochemical cell has proven a suitable method for decentralized analysis and screening [27]. Lithographic and printing techniques have been widely used to date in the fabrication of various wearable electrochemical sensors. Lithography represents a technique used for fabrication of integrated circuits and microchips that usually feature miniaturized size (from a few nanometers up to tens of millimeters), while printing methodologies embrace the reproduction of a pattern on a substrate by transferring material in the form of inks [28,29].

Herein, we report an extensive study regarding the effective on-site detection of Ent using commercial disposable screen-printed electrodes (SPEs) modified with an agar Ag/Au nanoalloy mixture (a semisolide electrolyte) with different possible applications such as early detection of *E. coli* in culture media, or in biological or environmental samples that are building bricks towards a rapid antibiogram, respectively decentralized detection devices. Two different culture media for Gram-negative bacteria growth, such as Drigalski Lactose Agar (DLA) and Miller Lysogeny broth (ML), were assessed to unravel the electrochemical fingerprint of Ent and evaluate any possible interferences with other bacterial metabolites. Furthermore, a simple decentralized setup for in-lab printed electrochemical cells was employed to detect the same target, this time by using flexible printing support, namely medical gloves. To our knowledge, there were no studies reported regarding the electrochemical detection of Ent as a marker of *E. coli* presence in culture media with water samples, and which use in-lab wearable sensors printed on a medical glove substrate. Such a development offers considerable promise for early rapid on-site detection of bacterial contamination.

## 2. Results and Discussion

The electrochemical fingerprint of Ent was assessed testing 100 μM solutions of the target analyte by differential pulse voltammetry (DPV) on different electrode configurations—namely SPE with graphite/gold/platinum-based working surface, glassy carbon electrodes (GCE), and custom in-lab printed electrodes with a graphite-based working surface—to assess the optimum electrode material (Figure 1).

Ent has in its structure three catecholamide groups linked by a trilactone ring. Based on previous studies where the electrochemical oxidation of catechol was observed at about 0.1–0.3 V on GCE [30,31], the electrochemical signal registered at approximatively 0.35 V was attributed to the electrochemical oxidation of the three catechol groups.

By far, the highest current intensity of approximatively 13 μA was observed on the graphite-based SPE at 0.05 V/Ag, while on GCE the characteristic peak was observed at about 0.27 V/Ag with a diminished current intensity. In this case, the difference between the electrodes’ surfaces (12.56 mm^2^ (SPE) vs. 7.065 mm^2^ (GCE)) needs to be taken into account. When the sensitivity of the electrodes’ surface was calculated (the values of the signal obtained divided by the concentration and by the area of the working electrode surface) similar results were obtained in this case, as can be seen in Table 1. The anodic shift of almost 0.2 V observed on GCE can be explained by the increased surface area of the SPE and the porosity of the surface. Thus, the improved electron transfer rate was reflected in both current intensity and peak potential. When assessed in the presence of Ent, the in-lab printed electrodes with the working surface designed with graphite-based conductive ink indicated an oxidation potential of 0.1 V/Ag/AgCl, while the current intensity was almost 6-fold diminished and the sensitivity of the electrodes’ surface, almost 10-fold in comparison with the SPEs. The results aligned with the expectations, because no pretreatment to improve their properties was included in the synthesis protocol. Overall, the fact that the detection of Ent was achieved with this type of electrodes represents a major advantage considering the further possibilities of customization, small production costs, and—more importantly—the versatility of the printing substrate, namely a medical glove.

On the Pt-based surface, the characteristic peak was registered at 0.16 V/Ag, but with a lower intensity than on graphite-based materials, while on the Au electrode the electrochemical oxidation of Ent was not clearly observed.

The variation of the peak position, current intensity, and sensitivity of the electrode surface on different electrode materials were summarized in Table 1.

Based on this knowledge, the detection of the target in culture media was further carried out on commercial SPEs with a graphite-based working surface. This represents an advantage from different points of view: (1) relatively low cost when reported to the ones with Au or Pt surfaces, and (2) potential for integration in portable devices for decentralized analysis, while the GCE could be used only in laboratory setup.

### 2.1. Results on the Commercial Graphite-Based SPEs

The working surface was modified either with agar with/without Au/Ag nanoalloy mixture (3:1 ratio) to assess the best configurations for fingerprinting this siderophore (Figure 2). A control experiment was performed in the absence of Ent and no anodic signal was observed. As expected, the anodic peak had almost the same intensity on the unmodified electrode and on the agar layer, proving that the electron transfer rate was not hindered by the drop-casted hydrogel. Subsequently, when the Au/Ag nanoalloy was mixed into the agar substrate, the current intensity was almost 4-fold higher due to the presence of noble metals’ particles that facilitated the electron transfer rate. The data are consistent with the results obtained on PyoC using this platform as shown in previous studies [32].

This configuration was further tested in the presence of various concentrations of standard solutions of Ent in 20 mM phosphate buffer saline (PBS) pH 7.4 (Figure 3a), as a preview for the assessment in specific culture media. The current intensity increased proportionally with the concentration from 1.25 to 37.5 µM, as can be seen in Figure 3b.

Unraveling the electrochemical behavior of the target in different bacterial growth media has the capacity to identify possible interferents and represents an important challenge towards the early detection of bacterial strains. The study describes the evaluation of Ent in (1) ML and (2) DLA. In both cases, the culture media was either diluted or suspended in 20 mM PBS pH 7.4, as described in Section 3.2.

ML is a nutrient-rich microbial broth that contains peptides, amino acids, water-soluble vitamins, and carbohydrates. ML broth allows fast growth and good growth yields for many species. It contains tryptone, yeast extract, NaCl, and distilled water. The pH 7 is adjusted with NaOH. The two organic ingredients are rich in oligopeptides and *E. coli* has several oligopeptide permeases and peptidases that allow the recovery of free amino acids from the oligopeptides [33].

The oxidation potential registered a slight anodic shift of about 0.05 V (Figure 4a) when compared to the data obtained only in PBS solution, certainly due to the presence of the peptides and amino acids which present electrochemical activity, as observed in the literature’s data [34]. The current intensity increased proportionally with the concentration of Ent from 6.25 to 37.5 µM, but the sensitivity was lower than the one obtained when using standard solutions in 20 mM PBS pH 7.4 (Figure 4b).

DLA is a selective and differential media for Gram-negative bacilli (*Enterobacteriaceae*). It is recommended for use with clinical specimens likely to contain mixed microbial flora—such as urine, sputum, or stool—because it allows a preliminary grouping of enteric and other Gram-negative bacilli. It contains agar, lactose as the source of carbon and fermentable carbohydrate, peptone, meat extract, yeast extract, sodium deoxycholate, crystal violet, and thiosulfate [35].

Following the suspending in PBS, the samples were spiked with different concentrations of Ent standard solutions and similar results with the ones from ML media were obtained. The same anodic shift of the oxidation potential (Figure 4c) and a slightly improved sensitivity were observed, as seen in Figure 4d.

For better comparison and systematization, the analytical parameters are depicted in Table 2.

As previewed, the sensitivity for the experiments in culture media was diminished, but this observation is not to be considered as an issue because the direct detection of Ent in specific medium for *E. coli* enables a new range of applications regarding the early detection of the bacteria, prior conventional incubation time, or the evolution of colonies based on the levels of the target.

### 2.2. In-Lab Printed Electrochemical Cells

Screen-printing involves printing conductive inks at a low pressure via a screen mesh/stencil containing a designed pattern which features a uniform thickness. The stencil can be made from a variety of materials including polyamide/polyester or stainless steel. To squeeze the conductive fluidic ink through the designed stencil, a metallic or rubber squeegee blade can be used. In this way, the ink is directly transferred to the desired substrate. In this case, the three electrodes were printed on a substrate—namely, the fingers of a medical glove (Figure 5a). The advantage of this approach is that the printed finger-based electrodes can withstand mechanical stress because of the soft and elastic properties of the medical glove, but also due to the flexible propriety of the insulating film.

The elaboration of the printed electrodes was documented step by step along the printing protocol. Firstly, the contacts and the reference electrode (RE) with Ag/AgCl conductive ink (Figure 5c) were printed—followed by printing of the counter electrodes (CE) and working electrodes (WE)—with carbon-based conductive ink (Figure 5d). After the insulation of the conductive wires with insulating polymer, the corresponding contacts for the WE, CE, and RE were connected to a miniaturized potentiostat that was operated via Bluetooth transmission by a smart tablet.

The electrodes were tested in the presence of Ent in PBS pH 7.4, in agar and in the presence of the Au/Ag nanoalloy as performed in the case of SPEs. The volume necessary to cover the three electrodes was established to be 60 μL due to the different configuration of the electrochemical cell that has 17 mm diameter, but the same agar:Au/Ag nanoalloy ratio was kept. In the presence of agar, the current intensity was about 0.26 μA registered at 0 V/Ag/AgCl, lower than the one on the unmodified electrode (Figure 6). This behavior was not observed on the commercial SPEs and could be associated to the electrodes’ graphite-based surface that was not submitted to any pretreatments. In the presence of the nanoalloy, a higher oxidation signal of about 1.1 μA at 0.13 V/Ag/AgCl was observed. As previewed, the presence of the Ag/Au nanoalloy determined an increase in the intensity of the oxidation peak (1.1 μA)—along with an anodic shift of 0.13 V—that could be linked to the surface porosity and to the presence of a polymer in the ink that was obtained after the polymerization of the monomer used for graphite particle immobilization onto the printing support. It is clearly visible that the overall current intensity of Ent oxidation peak is smaller than the values obtained on the graphite-based SPEs. The in-lab printed electrodes were not submitted to other procedures of pretreatment as in the case of commercial ones. This fact does not represent a major issue for the final purpose of the sensor, namely the rapid and early field detection of Ent.

The same protocol was applied to the electrodes printed on a thin flexible substrate and on the fingers of the medical glove, as depicted in Figure 7. This indicates that the novel printed electrodes have the capacity to successfully detect the target, even though the current intensity is significantly lower than on the commercially available SPEs. The anodic shift of the potential is caused by the surface porosity and the presence of a polymer in the ink (obtained after the polymerization of the monomer used for graphite particles immobilization onto the printing support). The smaller signal obtained for Ent on the printed electrodes could also be attributed to a high amount of insulating polymer in the composition of the ink for the in-lab printing that could influence the electrochemical behavior in the presence of the target. Moreover, the electrodes were used immediately after the elaboration protocol and no pretreatment was included in this approach.

The advantage of this approach lies in the possibility of optimization/customization of the composition of the ink, the small production cost, and—more importantly—the versatility of the printing technique that allows generating different wearable sensors depending on the desired applications.

The differences in current intensity are issued by the differences in the type of working surface area of in-lab printed WE (19.62 mm^2^) vs. graphite-based commercial SPEs (12.56 mm^2^). The electrochemical signal of Ent was registered at approximatively 0.13 V/Ag/AgCl, the data having a cathodic shift comparing with other carbon-based working surfaces that are listed in Table 1. The explanation lies also in the type of carbon-based conductive ink and the WE surface. For a better visualization, the analytical parameters of the tested sensors were presented in Table 3.

In the simultaneous assessment of PyoC and Ent (Figure 8), a cathodic shift for the peak position of Ent was observed but the intensity was almost the same.

The same trend regarding the current intensity was observed in the case of PyoC when both targets were found in the sample solution. This setup can be successfully used for the rapid detection of both markers and could indirectly and simultaneously prove the presence of *P. aeruginosa* and *E. coli*.

We agree that the method could—and will be—further improved to increase the sensitivity, but the goal of this study was to extend the studies performed on commercially available SPEs onto in-lab printed electrodes on an unconventional substrate, namely a medical glove. This achievement underlines the versatility of this strategy and the fact that the glove-printed electrochemical cells can be used for decentralized analysis when coupled with a miniaturized potentiostat and further reduce the costs invovled in hte acquisition of commercial electrodes.

Future studies will focus on the modification of the carbon ink to increase both specificity and sensitivity of the sensor. Furthermore, if possible, the electrochemical measurements will be correlated with the bacterial load to establish if this method could have a degree of certitude regarding the bacterial colonies development.

### 2.3. Evaluation of Ent Signal in Water Samples

The ability to detect bacteria markers in waters is a key feature for any analytical system as it would serve as an indicator for the presence of microbial contamination. The presence of Ent was monitored in real samples/complex matrixes: wastewater, groundwater, and tap water. Two types of samples collected from the wastewater treatment plant (primarily clarified wastewater and treated effluent) were spiked with 25 μM, 50 μM, and 75 μM of Ent.

The target could be detected in environmental samples, maintaining its fingerprint with its oxidation potential at graphite SPE surface kept between +0.12 V and +0.16 V.

Recovery percentages for each concentration were calculated based on the average value from three determinations, with reference to the average result being obtained from standard tests of the same concentration. Satisfactory recoveries (above 80%) were achieved on primarily clarified wastewater and treated effluent water for all the concentrations tested, which are represented in Figure 9.

Preliminary results in groundwater and tap water showed good recoveries for the 25 μM concentration (100.38% and 81.37%, respectively), as well as for 75 μM (99.52% and 108.89%, respectively), which is a promising basis for further tests. The presence of ions (chloride, calcium, magnesium etc.) in the treated water samples and other components in the raw untreated waters most likely have an influence on the signal of the analyte. All of the tests were performed in triplicate.

## 3. Materials and Methods

All the reagents were used as received, without any other pretreatment and purification. Ent, K_4_[Fe(CN)_6_], K_3_[Fe(CN)_6_], Na_2_HPO_4_, NaH_2_PO_4_, KCl, and ML were purchased from Sigma Aldrich (St. Louis, MO, USA). The Au/Ag nanoalloy suspension (citrate capped gold-silver nanoparticles with 40 nm diameter and maximum absorbance (Abs_max_) at 410 nm) was purchased from Nanovex (Oviedo, Spain), while the agar powder and DLA was provided by Merck (Darmstadt, Germany). All the solutions were prepared using Milli-Q ultrapure water (18 MΩ cm^−1^).

The SPEs with different working surfaces namely graphite (with a geometrical surface of 12.56 mm^2^), gold, and platinum were purchased from Metrohm Dropsens (Oviedo, Spain), while the GCE (with a geometrical surface of 7.065 mm^2^) were purchased from BASi (West Lafayette, IN, USA).

### 3.1. Generation of the Platform

The elaboration of the platform was performed following a published protocol developed by Cernat et al. [32]. Briefly, the three electrodes of the electrochemical cell were covered with a suspension of 0.5% (m/V) agar hydrogel in 20 mM PBS pH 7.4 and Au/Ag nanoalloy in a 3:1 ratio. The volume applied to generate the electrochemical cell in situ was established to be 30 μL and the electrode was kept at room temperature 2 min prior to any experiments to allow the formation of the semisolide layer.

### 3.2. Assessment of Enterobactin in Culture Media

The Ent samples were prepared in 20 mM PBS pH 7.4 in concentrations ranging from 0 to 100 μM. The volume samples were applied on the semisolide layer and allowed 3 min to homogenously diffuse. The electrochemical fingerprint of *E. coli* siderophore was registered by DPV in a potential range from −0.7 V to 1.0 V with a scan rate of 20 mV/s. Different electrode configurations were tested to assess the optimum electrochemical response for the target in terms of potential/current intensity.

The electrochemical behavior of Ent was evaluated using two different culture media for Gram-negative bacteria, namely DLA suspended in 20 mM PBS pH 7.4 (1 mg/0.05 mL PBS) and ML diluted 1:50 with 20 mM PBS pH 7.4. The electrochemical experiments were performed on an Autolab PGSTAT30.

### 3.3. Assessment of Enterobactin in Water Samples

The presence of Ent was evaluated in untreated and treated waters (wastewater, groundwater, and tap water). Wastewater was obtained from two collection tanks from the Someș sewage treatment plant: primarily clarified wastewater (without sludge or other pretreatment) and treated effluent. The wastewater samples were filtered through a syringe microfilter with 0.45 µM pore size prior to testing. Groundwater was collected from a well in Veștem village, Sibiu county. Tap water was collected from our research laboratory in Cluj-Napoca. The samples were stored in the fridge at 4 °C until analyses were performed.

No dilution or other pretreatments were applied to the water samples collected from the various sources. The samples were spiked with known concentrations of Ent (25 μM, 50 μM, and 75 μM), which were then tested using the same method described in Section 3.2.

### 3.4. Elaboration of the In-Lab Printed Electrochemical Cells

The in-lab elaborated electrochemical cells were printed either on a thin flexible polymeric substrate or on the fingers of a medical glove. The printing strategy was accomplished using AutoCAD (Autodesk, San Rafael, CA, USA) for fabrication of stainless steel through-hole 12″ × 12″ framed stencils with 150 μm thickness (Metal Etch Services, San Marcos, CA, USA). The stencils are characterized by well-defined and rigorous patterns for the design of the electrodes. Firstly, the stencil was located on top of the medical glove, covering the spots of the chosen fingers. A layer of the conductive inks was applied on the upper surface of the stencil. In the next step, a flexible rubber squeegee was dragging the inks over the stencil’s openings. Initially, the Ag/AgCl conductive layer was printed and dried on the plastic surface in an autoclave at 50 °C for 15 min. Subsequent to the complete drying of the Ag/AgCl conductive ink, the carbon conductive layer was printed, following the same drying procedure as for the Ag/AgCl ink. The RE is based on the Ag/AgCl conductive ink, while both WE and CE are designed by carbon conductive ink. Since the thickness of the stencil is 150 μm, the printing process on the sensing fingers ensures the same thickness of the ink layers. To decrease the shortcuts on the top of the connections, an insulator adhesive tape was attached on the surface of the electrodes. The non-conductive tape also presented the role of guaranteeing the dielectric separation of the three-electrode system, without damaging the active sensing area. The electrochemical sensors did not need any sort of pretreatment before their use.

The serpentine connections were also printed with the Ag/AgCl conductive ink. The silver-based serpentine connection presented a length of 6 cm for each finger and was planned to reach the base of each finger, where the conductive wires were attached. The conductive wires facilitated the connection between the medical glove and the potentiostat.

### 3.5. Assessment of Ent Using the Printed Electrodes on the Glove Substrate

After the printing on the laboratory glove was finished, the printed gloves were left overnight in an autoclave at 50 °C for complete drying of all ink layers. The glove with electrochemical sensors printed on one finger was then worn by one volunteer, positioned with the sensors facing upwards (see details in Figure 5). The conductive wires were carefully insulated and secured with adhesive tape along the glove so as to avoid any short-circuit or interference.

The corresponding contacts for the WE, CE, and RE were connected to a miniaturized potentiostat (EmStat Blue from PalmSens BV) which was operated with a specific software (PStouch from PalmSens BV) via Bluetooth by using a smart tablet (Samsung Galaxy mini). The software installed on the tablet ensured both the operation of the potentiostat and the storage of the recorded experimental data.

The three-electrode system printed on one glove finger was coated with a 60 μL drop of the agar-Au/Ag nanoalloy hydrogel. After 2 min, 20 μL sample solution containing the target analytes was casted (the final concentration of the target of 25 μM). After the 3 min necessary for the diffusion through the hydrogel layer, the measuring procedure was carried out by DPV with identical parameters as the one described in the case of the commercial SPEs.

The experiments carried out on the printed finger-based sensors employed 80 μL electrolyte (60 μL semisolide mixture and 20 μL sample solution), given the larger area of the electrochemical cell for the printed sensors (about 17 mm in diameter for the electrochemical cell) that needed to be covered compared to the surface area of the commercial SPEs (about 10 mm in diameter for the electrochemical cell).

The analytical signal of the target bacteria markers was recorded using a DPV testing procedure (Section 3.2).

## 4. Conclusions

The early detection of bacterial markers via electrochemical decentralized sensing devices could change the conventional protocols. Ent, a metabolite of *E. coli* that can be considered marker for this bacteria was for the first time electrochemically fingerprinted using two types of printed electrodes—namely, commercial SPEs and in-lab printed SPEs—in two distinct scenarios. The novelty of this study lies in the successful detection in specific culture media without major interferences from their composition, a prior step in an electrochemical antibiogram. The perspectives of the glove-based sensors are linked to the early detection of the bacteria prior the formation of the specific colonies, an important feature for healthcare practice. Furthermore, the presence of Ent was assessed in water samples with satisfactory results enabling new strategies for the monitoring water quality. This fact could be a turnover point for the detection of *E. coli* strains that are carrying genes involved in the mechanism of antibiotic resistance. Secondly, the in-lab setup using customizable printed electrodes on a medical glove—combined with a portable potentiostat paired to a smart device for wireless data transmission—allowed for the decentralized analysis of the target bacterial marker. This approach offers a perspective of early on-site detection of contamination which is very promising for relevant practical application.

## Figures and Tables

**Figure 1 ijms-23-09884-f001:**
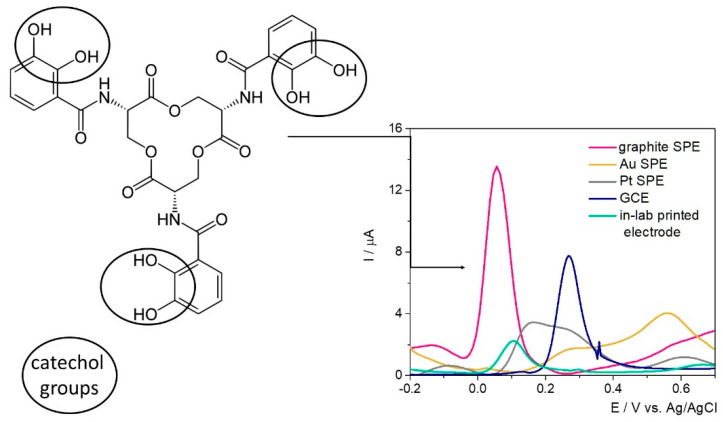
Ent structure and the electrochemical oxidation on different electrode materials.

**Figure 2 ijms-23-09884-f002:**
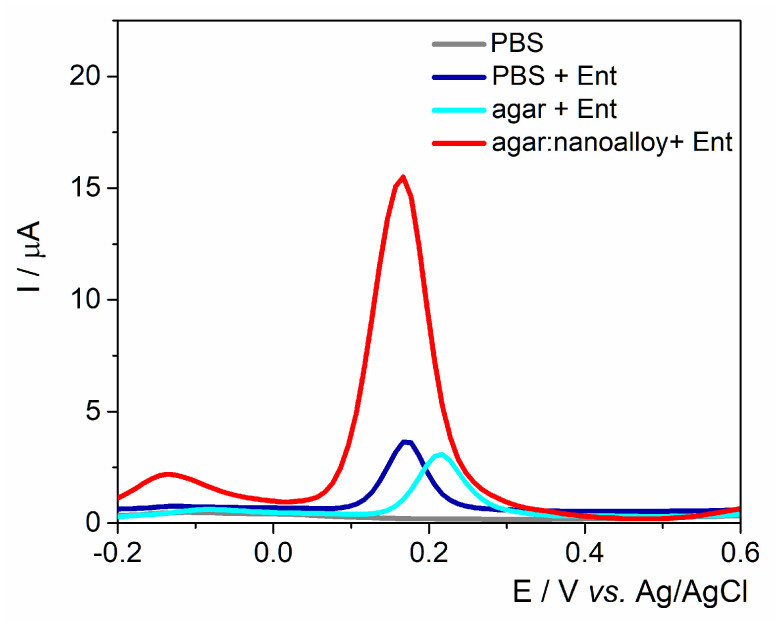
DPV for PBS pH 7.4 and 25 μM Ent on unmodified graphite SPE and modified with the agar hydrogel with/without the Au/Ag nanoalloy.

**Figure 3 ijms-23-09884-f003:**
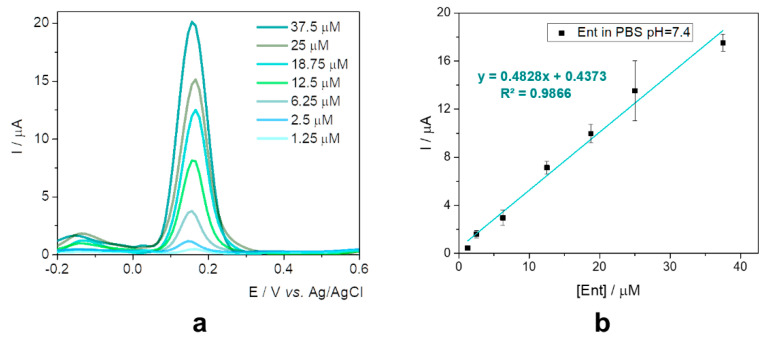
(**a**) DPVs for 1.25–37.5 μM Ent in PBS pH 7.4 at graphite SPE; (**b**) Calibration curve for Ent in PBS pH 7.4 with the corresponding equation and error bars (based on standard deviation for three tests performed on each concentration). LOD: 0.41 µM; LR: 1.25–37.5 µM.

**Figure 4 ijms-23-09884-f004:**
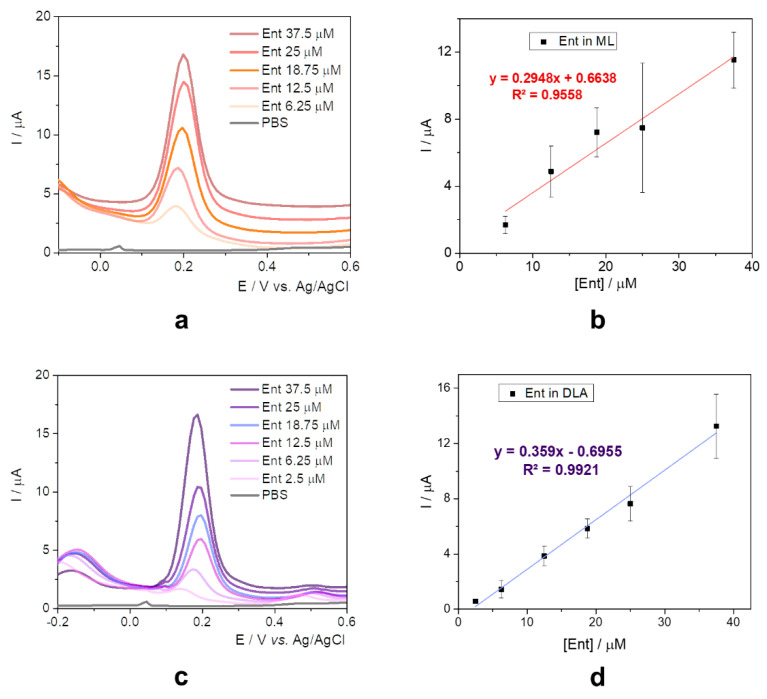
(**a**) DPVs for 6.25–37.5 μM Ent in ML diluted 1:50 with 20 mM PBS pH 7.4 at graphite SPE; (**b**) calibration curve for Ent in ML; (**c**) DPVs for 2.5–37.5 μM Ent in DLA suspended in 20 mM PBS pH 7.4 (1 mg/0.05 mL PBS) at graphite SPE; (**d**) calibration curve for Ent in DLA.

**Figure 5 ijms-23-09884-f005:**
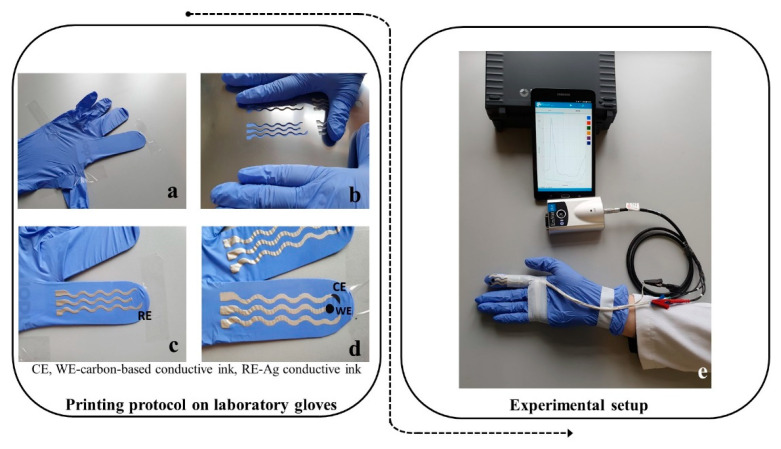
Images recorded during the printing protocol on laboratory gloves: (**a**) glove ready for printing; (**b**) stainless steel stencil with cut-outs for printing the contacts and the electrochemical cell; (**c**) contacts and RE printed on one glove finger; (**d**) electrochemical cell and contacts printed on a glove finger; (**e**) images with the printed glove wired for connection to the portable potentiostat and smart tablet that ensures the operation of the system via Bluetooth transmission and specific software.

**Figure 6 ijms-23-09884-f006:**
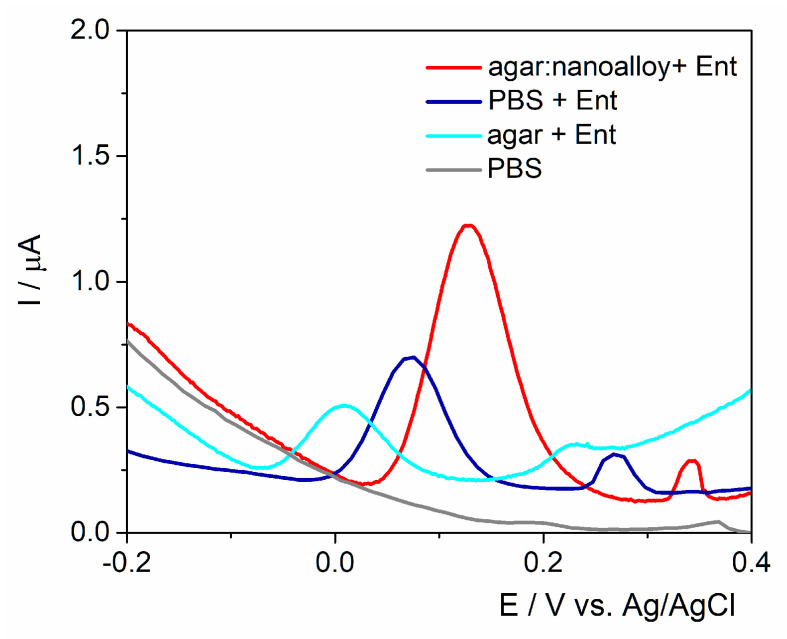
DPV for 25 μM Ent with an in-lab printed electrode on glove modified with 60 μL agar hydrogel without/with Au/Ag nanoalloy.

**Figure 7 ijms-23-09884-f007:**
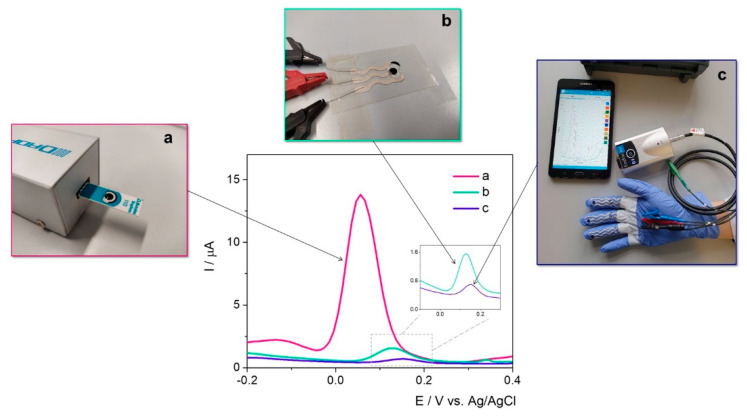
DPV for 25 μM Ent on agar:Au/Ag nanoalloy platform developed on (**a**) commercially available graphite SPE; (**b**) in-lab printed electrode on thin flexible substrate; (**c**) in-lab printed electrode on medical glove. Given the promising results obtained on graphite-based commercial SPEs for the detection of PyoC in previous studies [32] and Ent, this configuration was assessed for the individual, respectively simultaneous detection of both bacteria markers. In the presence of PyoC, a specific peak appeared at −0.4 V/Ag/AgCl consistent with prior published results [32].

**Figure 8 ijms-23-09884-f008:**
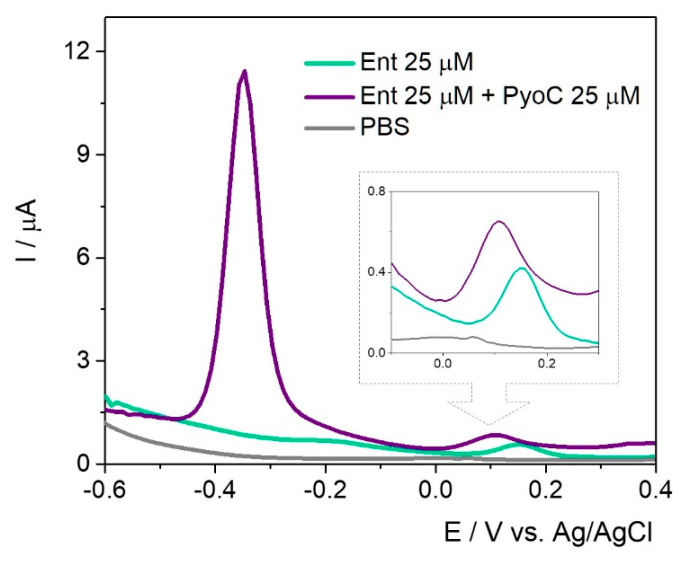
DPV for 20 mM PBS pH 7.4 (grey), 25 μM Ent (green), and 25 μM Ent + 25 μM PyoC (purple) on agar:Au/Ag nanoalloy platform developed on in-lab printed electrode on thin flexible substrate.

**Figure 9 ijms-23-09884-f009:**
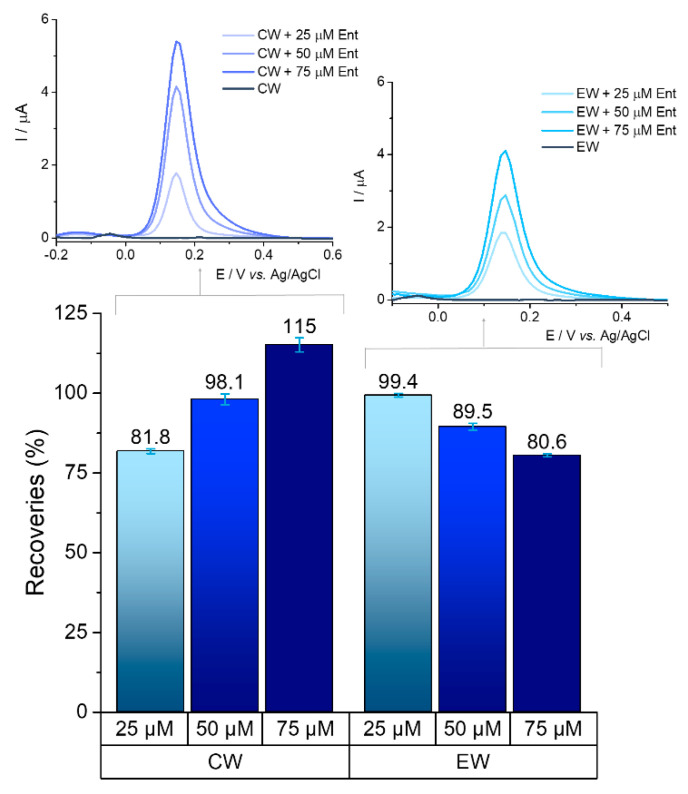
Recoveries obtained in wastewater samples spiked with three known concentrations of Ent; CW—primarily clarified wastewater (without sludge or other pretreatment) and EW—treated effluent; error bars represented based on standard deviation for three tests performed on each concentration; DPVs before (CW and EW) and after spiking with different concentrations of Ent (25 μM, 50 μM, and 75 μM).

**Table 1 ijms-23-09884-t001:** Analytical parameters regarding the electrochemical oxidation of Ent on different electrode materials.

Electrode Type	E/V	I/μA	S (A M^−1^ cm^−2^)
**Graphite SPEs**	**0.05**	**12.76**	**1.02**
GCE	0.27	7.49	1.06
**In-lab printed electrodes**	**0.10**	**2.02**	**0.10**
Au SPEs	0.56	2.22	0.18
Pt SPEs	0.16	3.26	0.26

**Table 2 ijms-23-09884-t002:** Analytical parameters for Ent detection in 20 mM PBS pH 7.4 and in specific culture media.

Media	E_ox_ (V)	Slope	Intercept	Linear Range (μM)	LOD (μM) (S/N = 3)
20 mM PBS pH 7.4	0.15	0.4828	0.4373	1.25–25	0.41
ML *	0.2	0.2948	0.6638	6.25–37.5	2.08
DLA **	0.2	0.3590	−0.6955	2.5–37.5	0.83

* diluted 1:50 with 20 mM PBS pH 7.4; ** suspended 1 mg/0.05 mL in 20 mM PBS pH 7.4.

**Table 3 ijms-23-09884-t003:** Analytical parameters for Ent detection in 20 mM PBS pH 7.4 and in the presence of PyoC using in-lab printed electrodes.

Method	Electrode	Modification Layer	Sample	Conc (µM)	E (V)	I (µA)	E (V)	I (µA)
					*PyoC*		*Ent*	
DPV	In-lab SPEs printed on plastic substrate	60 µL agar: Au/Ag nanoalloy	Ent	25	-	-	0.15	0.32
			PyoC + Ent	25 + 25	−0.34	10.39	0.11	0.38

## Data Availability

The experimental data are stored in the authors’ laboratory and can be consulted, if required, after a prior request to the corresponding author.

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
