# Peer review of "Proof of Concept for the Detection with Custom Printed Electrodes of Enterobactin as a Marker of *Escherichia coli"

_ijms, 2022, doi:10.3390/ijms23179884_

Round 1
Reviewer 1 Report
The present manuscript entitled “Proof of concept device for the detection with custom printed electrodes of Enterobactin (Ent) as a marker of Escherichia coli” is a very complete and interesting work in which the authors proposed the on-site detection of Ent using commercial disposable screen-printed electrodes modified with an agar Ag/Au nanoalloy mixture as semisolid electrolyte. Moreover, to increase the novelty, a custom configuration printed on a medical glove was employed to detect the target in the presence of another bacterial marker aiming for decentralized analysis when coupled with a miniaturized potentiostat. Analytical results are provided for wastewater and treated water samples collected from the municipal sewage treatment plant, groundwater, and tap water.
The present manuscript is very interesting and shows sufficient novelty and quality for being published in the IJMS journal, Special Issue Analytical Methods in Biomedical and Environmental Applications: A Molecular Approach. However, I recommend addressing the following minor suggestions before publishing:
1. It is mentioned from the beginning that enterobactin (Ent) is a siderophore specific to Escherichia coli. The authors may want to rephrase this sentence a bit since (in a quick search) it can be found also in other bacterial sp.: “is primarily found in Gram-negative bacteria, such as Escherichia coli and Salmonella typhimurium”. This might be taken into account for the statement made in lines 312-315 also, where it is pointed out that the detection of both biomarkers could determine the presence of P. aeruginosa and E. coli, please modify accordingly to also include other gram-negative bacterias such as Salmonella.
2. Could the authors add clarification for Figures 2 and 6? In both plots, the red fingerprint for “agar:NP+Ent” is mentioned, but NP is not explained in the text, by the context it can be understood that it is referring to the absence of nanoalloy, but it is not completely obvious, and could be confusing for the readers; just by adding in the caption between brackets (NP) should be enough.
3. Figure 3 is mentioned in the manuscript but both parts (A and B) are not, this might be required sometimes depending on the journal. Normally all figures and tables should be mentioned within the main text. For instance, Figures 4A and 4B are mentioned, but 4C and 4D are missing also.
4. Moreover, regarding the formatting of the figures, sometimes capital letters are used (A, B…) but others such as Figure 5 or Figure 7, show along the text, within the figure itself and also in the caption as “a, b, c…” and ideally this details must be unified in the entire document if possible.
5. When both culture media are introduced, DLA is very well explained but ML is plainer, could the authors add a similar paragraph for it (with more characteristics about its use and sample affinities)?
6. Interestingly, the authors mentioned in lines 286-290 that the “novel printed electrodes have the capacity to successfully detect the target, even though the current intensity is significantly lower than on the commercially available SPEs”. Although an explanation for the shift is provided just immediately, the explanation for the decrease in the intensity of the signal is missing yet. It is only in line 302 where this explanation is added. I believe that at least a short explanation could be facilitated once it is mentioned in line 286 since it is significantly pointed out in the sentence. As expected, the reason for the decrease was likely due to the active surface area of the working electrodes, however, only 2 areas are provided in lines 303-304 while 3 different signals/decreases are shown in Figure 7. Could the authors explain why or clarify this? In the caption of Figure 7 three options are shown: “a. commercially available graphite SPE; b. in-lab printed electrode on thin flexible substrate; c. in-lab printed electrode on medical glove. It is understood then that only two diameters are present (19.62 and 12.56 mm2), but then, why there are differences between the two “in-lab printed” electrodes? And also, I think there is a mistake in lines 303 and 304 where the biggest signal (at least following what appears in Figure 7) is coming from the commercial electrode, hence, the biggest area should be assigned to the commercial one, right now it is assigned to the “in-lab” one.
7. The paragraph starting in line 348 describes the recoveries obtained after measuring the preliminary results in groundwater and tap water, it is a bit confusing to have in Figure 9 (caption included) other titles for those samples “CW - primarily clarified wastewater (without sludge or other pretreatment) and EW - treated effluent” and also the numbers shown in the paragraph are not corresponding with the ones that the Figure 9 shows. If this is a mistake I suggest correcting it, if it is not, I think it can be confusing and clarifications must be added to facilitate the understanding of the story.
Overall, I think the present manuscript exhibits high quality for being published in the IJMS journal, only minor modifications are required.
Author Response
Answers for Reviewer 1:
We thank the reviewer for reading our manuscript and for the comments that helped us to improve its quality.
All observations were carefully considered, and we tried to answer punctually.
All changes and corrections in the manuscript are marked with Track Changes.
The present manuscript entitled “Proof of concept device for the detection with custom printed electrodes of Enterobactin (Ent) as a marker of Escherichia coli” is a very complete and interesting work in which the authors proposed the on-site detection of Ent using commercial disposable screen-printed electrodes modified with an agar Ag/Au nanoalloy mixture as semisolid electrolyte. Moreover, to increase the novelty, a custom configuration printed on a medical glove was employed to detect the target in the presence of another bacterial marker aiming for decentralized analysis when coupled with a miniaturized potentiostat. Analytical results are provided for wastewater and treated water samples collected from the municipal sewage treatment plant, groundwater, and tap water.
Answer: We thank the reviewer for the appreciations related to our manuscript.
The present manuscript is very interesting and shows sufficient novelty and quality for being published in the IJMS journal, Special Issue Analytical Methods in Biomedical and Environmental Applications: A Molecular Approach. However, I recommend addressing the following minor suggestions before publishing:
- It is mentioned from the beginning that enterobactin (Ent) is a siderophore specific to Escherichia coli. The authors may want to rephrase this sentence a bit since (in a quick search) it can be found also in other bacterial sp.: “is primarily found in Gram-negative bacteria, such as Escherichia coli and Salmonella typhimurium”. This might be taken into account for the statement made in lines 312-315 also, where it is pointed out that the detection of both biomarkers could determine the presence of P. aeruginosa and E. coli, please modify accordingly to also include other gram-negative bacterias such as Salmonella.
Answer: Indeed, Enterobactin is synthesized by other G- bacteria, but in majority of the studies it was found to be as a marker of E. coli.
However, we replaced in the manuscript this affirmation by adding the following sentence: „or other Gram-negative bacteria”.
- Could the authors add clarification for Figures 2 and 6? In both plots, the red fingerprint for “agar:NP+Ent” is mentioned, but NP is not explained in the text, by the context it can be understood that it is referring to the absence of nanoalloy, but it is not completely obvious, and could be confusing for the readers; just by adding in the caption between brackets (NP) should be enough.
Answer: The figures were modified accordingly by inserting nanoalloy instead of NP.
- Figure 3 is mentioned in the manuscript but both parts (A and B) are not, this might be required sometimes depending on the journal. Normally all figures and tables should be mentioned within the main text. For instance, Figures 4A and 4B are mentioned, but 4C and 4D are missing also.
Answer: The suggested modifications were applied in the revised version of the manuscript.
- Moreover, regarding the formatting of the figures, sometimes capital letters are used (A, B…) but others such as Figure 5 or Figure 7, show along the text, within the figure itself and also in the caption as “a, b, c…” and ideally this details must be unified in the entire document if possible.
Answer: The capital letters in Figures 3 and 4 were replaced with small letters.
- When both culture media are introduced, DLA is very well explained but ML is plainer, could the authors add a similar paragraph for it (with more characteristics about its use and sample affinities)?
Answer: The following section was inserted in the main manuscript:
“ML broth allows fast growth and good growth yields for many species. It contains tryptone, yeast extract, NaCl, and distilled water. The pH 7 is adjusted with NaOH. The two organic ingredients are rich in oligopeptides and E. coli has several oligopeptide permeases and peptidases that allow the recovery of free amino acids from the oligopeptides. (Escherichia coli Physiology in Luria-Bertani Broth doi: 10.1128/JB.01368-07)”
- Interestingly, the authors mentioned in lines 286-290 that the “novel printed electrodes have the capacity to successfully detect the target, even though the current intensity is significantly lower than on the commercially available SPEs”. Although an explanation for the shift is provided just immediately, the explanation for the decrease in the intensity of the signal is missing yet. It is only in line 302 where this explanation is added. I believe that at least a short explanation could be facilitated once it is mentioned in line 286 since it is significantly pointed out in the sentence. As expected, the reason for the decrease was likely due to the active surface area of the working electrodes, however, only 2 areas are provided in lines 303-304 while 3 different signals/decreases are shown in Figure 7. Could the authors explain why or clarify this? In the caption of Figure 7 three options are shown: “a. commercially available graphite SPE; b. in-lab printed electrode on thin flexible substrate; c. in-lab printed electrode on medical glove. It is understood then that only two diameters are present (19.62 and 12.56 mm2), but then, why there are differences between the two “in-lab printed” electrodes? And also, I think there is a mistake in lines 303 and 304 where the biggest signal (at least following what appears in Figure 7) is coming from the commercial electrode, hence, the biggest area should be assigned to the commercial one, right now it is assigned to the “in-lab” one.
Answer: The geometrical surface area of the commercial SPE is 12.56 mm2 while of the in-lab printed is larger, of 19.62 mm2. Indeed, we expected to achieve a higher signal on the in-lab printed electrodes based on this fact. The explanation for the lower signal obtained lies in the type of the surface and not in the geometrical area. Moreover, there is a difference between the conductive inks and the fact that we used the electrode right after the elaboration without any pretreatment determined a smaller signal. Regarding the two different signals on in-lab printed electrodes, the smaller one is obtained on the electrode printed on the finger of a medical glove, and the nature of the substrate is possible to influence the electrochemical signal. Further studies will be carried on when Ent will be assessed in biological samples, depending on the type of substrate selected for the application. The higher signal was obtained on the commercial SPE, and that is the reason that the water samples were tested on this type of electrode.
The following sentence was inserted in the manuscript:
„The smaller signal obtained for Ent on the printed electrodes could also be attributed to high amount of insulating polymer in the composition of the ink for the in-lab printed that could influence the electrochemical behaviour in the presence of the target. Moreover, the electrodes were used immediately after the elaboration protocol and no pretreatment was included in this approach.”
- The paragraph starting in line 348 describes the recoveries obtained after measuring the preliminary results in groundwater and tap water, it is a bit confusing to have in Figure 9 (caption included) other titles for those samples “CW - primarily clarified wastewater (without sludge or other pretreatment) and EW - treated effluent” and also the numbers shown in the paragraph are not corresponding with the ones that the Figure 9 shows. If this is a mistake I suggest correcting it, if it is not, I think it can be confusing and clarifications must be added to facilitate the understanding of the story.
Answer: The Ent presence was tested in different types of water samples: tap, ground and wastewater (primarily clarified wastewater and treated effluent). Figure 9 presents the data obtained on wastewater samples, but in the text are mentioned the recoveries obtained for both tap and groundwater. The percentages in the text refer to the tap and groundwater samples that are not shown in Figure 9.
Overall, I think the present manuscript exhibits high quality for being published in the IJMS journal, only minor modifications are required.
Reviewer 2 Report
After carefully reading the manuscript I can say that the manuscript is well-structured and presents information with a certain degree of novelty. The obtained results are compared with literature data and are supported by evidences, explanations and correlations being provided.
However, I have some comments and recommendations:
- page 2, line 68: please replace “methods of analysis”
- page 4, lines 153-155, Table 1 and page 9, lines 302-304: for a better characterization of the electrode material and significant comparisons, the parameter sensitivity must be calculated.
- page 5, Figure 3B: please, explain the large error bars obtained for 25 μM ENT. Moreover, it is not obvious from Figure 3B if the point from 37.5 μM ENT was taken into account in order to establish the linear range. Please, modify the figure in order to highlight the determined linear range, namely 1.25-25 μM ENT. R2 = 0.9866 (which is rather small) and the presented equation characterize the dependence that includes the point from 37.5 μM ENT?
- page 6, lines 209-211: please, provide a possible explanation
- page 6, Figure 4B: R2 is too small. Can the authors explain why the error bars are so large, especially for 25 μM ENT?
- page 9, lines 309-311: please, specify what "slight increase" means.
- page 10, lines 345-347: can these percentage recovery values (between 80.6 and 115 %) be considered acceptable for this concentration level? Please, provide some references.
-page 10, Section 2.3 Evaluation of ENT signal in water samples: why the water samples were not analyzed using the in-lab printed electrodes on medical glove?
- page 11, Figure 9: it would be more relevant if, instead of the obtained recoveries, Figure 9 would contain some voltammograms registered for the water samples, before and after spiking.
- page 13, lines 450-465: the conclusions must be reformulated / completed due to the fact that they do not sufficiently highlight the originality and the significant contributions of the study.
- the study should include details related to the surface characterization of the modified electrodes and also to their selectivity.
In conclusion, the manuscript needs major revision before publication could be recommended.
Author Response
Answers for Reviewer 2:
We thank the reviewer for reading our manuscript and for the comments that helped us to improve its quality.
All observations were carefully considered, and we tried to answer punctually.
All changes and corrections in the manuscript are marked with Track Changes.
After carefully reading the manuscript I can say that the manuscript is well-structured and presents information with a certain degree of novelty. The obtained results are compared with literature data and are supported by evidences, explanations and correlations being provided.
Answer: We thank the reviewer for the appreciations related to our manuscript, thank the reviewer for all the comments and suggestions. We revised the manuscript considering the recommendations we received.
However, I have some comments and recommendations:
- page 2, line 68: please replace “methods of analysis”
Answer: This line was rephrased as “the most important microbiological parameters that should be determined”.
- page 4, lines 153-155, Table 1 and page 9, lines 302-304: for a better characterization of the electrode material and significant comparisons, the parameter sensitivity must be calculated.
Answer: For the selection of the electrode material, no calibration curves were drawn for all the tested materials, so the sensitivity is not available to be included in Table 1 as suggested by the reviewer. In each individual case, only one concentration was tested for the enterobactin solution, and the electrode material that generated the highest value of the oxidation current intensity was selected for the development of the sensor. We tried to optimize the use of Ent standard and we performed the experiments only on the surface with the higher signal.
- page 5, Figure 3B: please, explain the large error bars obtained for 25 μM ENT. Moreover, it is not obvious from Figure 3B if the point from 37.5 μM ENT was taken into account in order to establish the linear range. Please, modify the figure in order to highlight the determined linear range, namely 1.25-25 μM ENT. R2 = 0.9866 (which is rather small) and the presented equation characterize the dependence that includes the point from 37.5 μM ENT?
Answer: Although the printed planar electrodes used in the study are commercial, purchased from Dropsens, a certain interelectrodic variability (below 5%) was observed. Also, Ent detection is done with the help of a semi-solid gel film containing an alloy of Au and Ag nanoparticles. When modifying the electrodes with gel, a rather viscous suspension was manipulated, thus, some differences may appear between the actual volume of the drop deposited on the electrode, but also in the content of alloy nanoparticles in each drop. All this can lead to inter-electrode variability and thus these errors were obtained, just by chance they are higher at the concentration of 25 μM ENT. Taking into account all these aspects, we consider that the errors obtained as an average from at least 3 determinations on each concentration (meaning at least 3 different sensors) as well as R2 are acceptable.
Sorry for this error, the equation included the 37.5 μM concentration, and the corresponding DPV was inserted in Figure 3a.
- page 6, lines 209-211: please, provide a possible explanation
Answer: A possible n explanation was provided in the manuscript as follows:
“The oxidation potential registered a slight anodic shift of about 0.05 V when reported to the data obtained only in PBS solution, certainly due to the presence of the peptides and amino acids which present electrochemical activity (https://www.ncbi.nlm.nih.gov/pmc/articles/PMC8699811/ ) (Figure 4a).”
- page 6, Figure 4B: R2 is too small. Can the authors explain why the error bars are so large, especially for 25 μM ENT?
Answer: Although the printed planar electrodes used in the study are commercial, purchased from Dropsens, a certain interelectrodic variability (below 5%) was observed. Also, Ent detection is done with the help of a semi-solid gel film containing an alloy of Au and Ag nanoparticles. When modifying the electrodes with gel, a rather viscous suspension was manipulated, thus, some differences may appear between the actual volume of the drop deposited on the electrode, but also in the content of alloy nanoparticles in each drop. All this can lead to inter-electrode variability and thus these errors were obtained, just by chance they are higher at the concentration of 25 μM ENT. Taking into account all these aspects, we consider that the errors obtained as an average from at least 3 determinations on each concentration (meaning at least 3 different sensors) as well as R2 are acceptable.
- page 9, lines 309-311: please, specify what "slight increase" means.
Answer: This was a mistake that has been revised.
- page 10, lines 345-347: can these percentage recovery values (between 80.6 and 115 %) be considered acceptable for this concentration level? Please, provide some references.
Answer: We cannot correlate the quantity of Ent with the E. coli colonies and, to our knowledge, there are no studies regarding this topic. Further studies are previewed to assess the link between quantity of Ent and bacteria’s colonies. However, we presented some regulations in the Introduction section (lines 63-77) where no bacteria should be found in drinking water. Based on these aspects we considered arbitrary that a recovery between 80-120 % would be suitable to assess the water quality. In this case, even a YES/NO test should provide enough reasons for the water samples to be further tested.
-page 10, Section 2.3 Evaluation of ENT signal in water samples: why the water samples were not analyzed using the in-lab printed electrodes on medical glove?
Answer: The electrochemical signal of standard solutions of Ent was lower in the case of in-lab printed electrodes and we attributed it to the fact that the electroactive surface was not submitted to any pretreatment after the printing protocol. In this case, to simplify and to achieve better recoveries, the commercial SPEs were used. Moreover, the use of in-lab printed electrodes was only a suggested application and further studies will be carried on when Ent will be assessed in biological samples, depending on the type of substrate selected for the application. The higher signal was obtained on the commercial SPE, and that is the reason that the water samples were tested on this type of electrode.
- page 11, Figure 9: it would be more relevant if, instead of the obtained recoveries, Figure 9 would contain some voltammograms registered for the water samples, before and after spiking.
Answer: Figure 9 was modified according to the indication of the reviewer.
- page 13, lines 450-465: the conclusions must be reformulated / completed due to the fact that they do not sufficiently highlight the originality and the significant contributions of the study.
Answer: The modifications were made as suggested by the reviewer in the conclusions section.
- the study should include details related to the surface characterization of the modified electrodes and also to their selectivity.
Answer: Both electrochemical and surface characterization studies were done before on the graphite-based SPE modified with the agar-Au/Ag alloy in the study for the detection of Pyocyanin (doi:10.1007/s00216-019-01857-4), thus we did not find it necessary to repeat it, and we focused more on analysing the real samples.
In conclusion, the manuscript needs major revision before publication could be recommended.
Reviewer 3 Report
The authors report a promising approach for the detection of enterobactin in environmental samples with custom printed electrodes.
I recommend that during manuscript revision, authors should consult a native English speaker in order to correct the word order in several sentences, e.g. lines 309 – 311: In the simultaneous assessment of PyoC and Ent, a cathodic shift was observed. There are several more examples in the text where similar corrections would need to be made (e.g. line 153-155).
Title
Lines 2-3, Proof of concept for the detection of enterobactin as a marker of E. coli using custom printed electrodes
Abstract
Line 18, modulate? What is meant here, e.g. adapt?
Lines 19-21, the sentences needs to be rephrased for clarity (“is prone to”); what is the real target, new methods (method principles), new selective markers for certain pathogens, or both?
Introduction
Line 36, what is the meaning of “did not pass in a secondary plan”?
Line 75, parametric value?
Line 82, major disadvantage is the long analysis time and
Lien 106, what is meant with “rephrase these strategies”? Which strategies? Shall they be replaced, improved, other?
Results and discussion
Lines 153-155, … differences need to be taken into account
Table 1, the current signals according to Figure 1 are ca. 13.5 microA for the graphite SPE and ca. 4 microA for the Au SPE
Figure 2, what is NP?
Line 220, the sensitivity as deduced from the calibration lines in Figures 4B and D is almost the same.
Figure 4B, the linearity is very poor, can this be explained by the (only) few replicates analysed or can this be attributed to the nature of the matrix (e.g. potentially higher inhomogeneity of ML versus DLA)?
Line 227, … are depicted in Table 2
Lines 229-233, it is not really clear from the text why reduced sensitivity “is not considered an issue”
Line 246, because of
Line 267, 0.26 microA, Figure 6 shows ca. 0.5 microA
Line 271, 1.1 microA, Figure 6 shows ca. 1.25 microA
Line 351, chloride?
Figure 9, error bars showed be shown to reflect the variability of results
Materials and Methods
Line 363, city and country of providers missing
Liner 392, pore size?
Line 401, printed on several fingers?
Line 405, characterized by well-defined
Line 409, Initially, the layer was printed
Line 410, Subsequent to
Line 411, … the carbon conductive layer was printed
Author Response
Answer for Reviewer 3:
We thank the reviewer for reading our manuscript and for the comments that helped us to improve its quality.
All observations were carefully considered, and we tried to answer punctually.
All changes and corrections in the manuscript are marked with Track Changes.
The authors report a promising approach for the detection of enterobactin in environmental samples with custom printed electrodes.
I recommend that during manuscript revision, authors should consult a native English speaker in order to correct the word order in several sentences, e.g. lines 309 – 311: In the simultaneous assessment of PyoC and Ent, a cathodic shift was observed. There are several more examples in the text where similar corrections would need to be made (e.g. line 153-155).
Answer: The manuscript was thoroughly revised according to the suggestions of the reviewer and the language and grammar errors were corrected.
Title
Lines 2-3, Proof of concept for the detection of enterobactin as a marker of E. coli using custom printed electrodes
Answer: The title was modified as suggested.
Abstract
Line 18, modulate? What is meant here, e.g. adapt?
Answer: Yes, that is correct. It is means as adjust, adapt. The term was replaced in the main manuscript with improve.
Lines 19-21, the sentences needs to be rephrased for clarity (“is prone to”); what is the real target, new methods (method principles), new selective markers for certain pathogens, or both?
Answer: The direct detection of bacteria via metabolites represents a new method for bacteria detection. The electrochemical detection of E. coli via enterobactin was reported for the first time in this study to our knowledge.
The sentence in the manuscript was replaced with:
“Identifying new markers and analysis methods represents an attractive strategy for the simple and rapid of the pathogen and has the potential to replace the existing enumeration-based/time-consuming methods”
Introduction
Line 36, what is the meaning of “did not pass in a secondary plan”?
Answer: Despite the research on COVID-19, the detection of bacteria remained a main research topic.
Line 75, parametric value?
Answer: The word “parametric” was deleted.
Line 82, major disadvantage is the long analysis time and
Answer: The following sentence was inserted in the manuscript “The major disadvantage is represented by the long timeframe”.
Lien 106, what is meant with “rephrase these strategies”? Which strategies? Shall they be replaced, improved, other?
Answer: the strategies are represented by the conventional methods of bacteria detection, namely the culturing plates and enumeration methods. The term was replaced by “reshape”. It means that new rapid detection device could help rethink the actual protocols.
Results and discussion
Lines 153-155, … differences need to be taken into account
Answer: The sentence was revised after the suggestion.
Table 1, the current signals according to Figure 1 are ca. 13.5 microA for the graphite SPE and ca. 4 microA for the Au SPE
Answer: The peak intensity values mentioned in the manuscript were double checked and are correct. The peak intensities were calculated using NOVA features, namely peak search on the raw data (from the baseline to the top of the peak). The DPVs were presented without any translation of the data and they don’t have the same baseline.
Figure 2, what is NP?
Answer: The term “NP” has been replaced with “nanoalloy”.
Line 220, the sensitivity as deduced from the calibration lines in Figures 4B and D is almost the same.
Answer: Yes, the observation is true. A small difference in sensitivity for ENT detection in two different culture media is beneficial, demonstrating the small matrix effect given by the compositional differences between the two media.
Figure 4B, the linearity is very poor, can this be explained by the (only) few replicates analysed or can this be attributed to the nature of the matrix (e.g. potentially higher inhomogeneity of ML versus DLA)?
Answer: Although the printed planar electrodes used in the study are commercial, purchased from Dropsens, a certain interelectrodic variability (below 5%) was observed. Also, Ent detection is done with the help of a semi-solid gel film containing an alloy of Au and Ag nanoparticles. When modifying the electrodes with gel, a rather viscous suspension was manipulated, thus, some differences may appear between the actual volume of the drop deposited on the electrode, but also in the content of alloy nanoparticles in each drop. All this can lead to inter-electrode variability and thus these errors were obtained, just by chance they are higher at the concentration of 25 μM ENT. Taking into account all these aspects, we consider that the errors obtained as an average from at least 3 determinations on each concentration (meaning at least 3 different sensors) as well as R2 are acceptable.
Line 227, … are depicted in Table 2
Answer: The sentence was revised after the suggestion.
Lines 229-233, it is not really clear from the text why reduced sensitivity “is not considered an issue”
Answer: The detection of Ent can be correlated to the presence of E. coli, thus even if the sensitivity is not excellent, it signals the presence of bacteria and enables future measures to be implemented. Further studies will be carried on in order to adapt the surface of the in-lab printed electrodes for better sensitivity.
Line 246, because of
Answer: It was modified in the main manuscript.
Line 267, 0.26 microA, Figure 6 shows ca. 0.5 microA
Line 271, 1.1 microA, Figure 6 shows ca. 1.25 microA
Answer: The peak intensity values mentioned in the manuscript were double checked and are correct. The peak intensities were calculated using NOVA features, namely peak search on the raw data (from the baseline to the top of the peak). The DPVs were presented without any translation of the data and they don’t have the same baseline.
Line 351, chloride?
Answer: The misspelling was corrected.
Figure 9, error bars showed be shown to reflect the variability of results
Answer: Figure 9 was modified accordingly, error bars and DPV plots were added to better reflect the results.
Materials and Methods
Line 363, city and country of providers missing
Answer: The missing information was included in the revised version of the manuscript.
Liner 392, pore size?
Answer: The mistake was corrected.
Line 401, printed on several fingers?
Answer: The sentence was revised and the missing information was added.
Line 405, characterized by well-defined
Line 409, Initially, the layer was printed
Line 410, Subsequent to
Line 411, … the carbon conductive layer was printed
Answer: The modifications were done in the main text.
Round 2
Reviewer 2 Report
After reading the revised manuscript and the responses given by the authors, I can say that the manuscript was improved. The authors took into account several recommendations, making various changes and providing some information and explanations.
However, I still have some comments / observations:
- page 4, lines 153-155, Table 1 and page 9, lines 302-304: for a better characterization of the electrode material and significant comparisons, the parameter sensitivity must be calculated.
Answer: For the selection of the electrode material, no calibration curves were drawn for all the tested materials, so the sensitivity is not available to be included in Table 1 as suggested by the reviewer. In each individual case, only one concentration was tested for the enterobactin solution, and the electrode material that generated the highest value of the oxidation current intensity was selected for the development of the sensor. We tried to optimize the use of Ent standard and we performed the experiments only on the surface with the higher signal.
I did not refer to the slope of the equation that characterizes the calibration curve, but to the sensitivity of the electrode surface (S expressed as A/((mol/L)·cm2), a parameter calculated by dividing the value of the signal obtained (i) by the concentration (c) and by the area of the working electrode surface (A): S = i/(c*A). It is obvious that a higher signal will be obtained on a larger electrode surface area. For this reason, for a significant comparison of different electrode surfaces the sensitivity is useful.
- page 6, lines 209-211: please, provide a possible explanation
Answer: A possible n explanation was provided in the manuscript as follows: “The oxidation potential registered a slight anodic shift of about 0.05 V when reported to the data obtained only in PBS solution, certainly due to the presence of the peptides and amino acids which present electrochemical activity (https://www.ncbi.nlm.nih.gov/pmc/articles/PMC8699811/ ) (Figure 4a).”
The paragraph referred to was at lines 209-2011 in the initial manuscript, in the revised manuscript being at lines 218-220, namely: “The current intensity increased proportionally with the concentration of Ent from 6.25 to 37.5 μ M, but the sensitivity was lower than the one obtained when using standard solutions in 20 mM PBS pH 7.4 (Figure 4b).” The required explanation it was related to the lower sensitivity obtained.
-page 10, Section 2.3 Evaluation of ENT signal in water samples: why the water samples were not analyzed using the in-lab printed electrodes on medical glove?
Answer: The electrochemical signal of standard solutions of Ent was lower in the case of in-lab printed electrodes and we attributed it to the fact that the electroactive surface was not submitted to any pretreatment after the printing protocol. In this case, to simplify and to achieve better recoveries, the commercial SPEs were used. Moreover, the use of in-lab printed electrodes was only a suggested application and further studies will be carried on when Ent will be assessed in biological samples, depending on the type of substrate selected for the application. The higher signal was obtained on the commercial SPE, and that is the reason that the water samples were tested on this type of electrode.
Taking into consideration the answer given by the authors, the water samples analysis using also the in-lab printed electrodes on medical glove would be even more significant, the comparison of the results being the basis of the proof of these electrodes applicability.
- page 11, Figure 9: it would be more relevant if, instead of the obtained recoveries, Figure 9 would contain some voltammograms registered for the water samples, before and after spiking.
Answer: Figure 9 was modified according to the indication of the reviewer.
The differential pulse voltammograms named CW_blank and EW_blank are the voltammograms registered for the water samples before spiking and the other are those obtained after spiking with 25 μM, 50 μM and 75 μM Ent?
In conclusion, my opinion is that the manuscript needs minor revision.
Author Response
Answers for Reviewer 2:
We thank the reviewer for reading our revised manuscript and for the supplementary observations. The feedback provided helped us make adjustments which, we hope, clarify better our work and results.
We answered and explained punctually to the comments given.
After reading the revised manuscript and the responses given by the authors, I can say that the manuscript was improved. The authors took into account several recommendations, making various changes and providing some information and explanations.
However, I still have some comments / observations:
- page 4, lines 153-155, Table 1 and page 9, lines 302-304: for a better characterization of the electrode material and significant comparisons, the parameter sensitivity must be calculated.
Answer: For the selection of the electrode material, no calibration curves were drawn for all the tested materials, so the sensitivity is not available to be included in Table 1 as suggested by the reviewer. In each individual case, only one concentration was tested for the enterobactin solution, and the electrode material that generated the highest value of the oxidation current intensity was selected for the development of the sensor. We tried to optimize the use of Ent standard and we performed the experiments only on the surface with the higher signal.
I did not refer to the slope of the equation that characterizes the calibration curve, but to the sensitivity of the electrode surface (S expressed as A/((mol/L)·cm2), a parameter calculated by dividing the value of the signal obtained (i) by the concentration (c) and by the area of the working electrode surface (A): S = i/(c*A). It is obvious that a higher signal will be obtained on a larger electrode surface area. For this reason, for a significant comparison of different electrode surfaces the sensitivity is useful.
Answer: We calculated the sensitivity for the different electrode surfaces as suggested by the reviewer and now this parameter has been added in a separate column in Table 1 in the revised manuscript.
- page 6, lines 209-211: please, provide a possible explanation
Answer: A possible n explanation was provided in the manuscript as follows: “The oxidation potential registered a slight anodic shift of about 0.05 V when reported to the data obtained only in PBS solution, certainly due to the presence of the peptides and amino acids which present electrochemical activity (https://www.ncbi.nlm.nih.gov/pmc/articles/PMC8699811/ ) (Figure 4a).”
The paragraph referred to was at lines 209-2011 in the initial manuscript, in the revised manuscript being at lines 218-220, namely: “The current intensity increased proportionally with the concentration of Ent from 6.25 to 37.5 μM, but the sensitivity was lower than the one obtained when using standard solutions in 20 mM PBS pH 7.4 (Figure 4b).” The required explanation it was related to the lower sensitivity obtained.
Answer: The tests in culture media samples and standard solutions in PBS were performed using the same platform (commercial graphite SPEs modified with the agar:nanoalloy mixture), which means having the same type of electrode surface. In the revised manuscript, in the quoted lines we are comparing the sensitivities of the method which refer to the slopes obtained for the calibration curves determined in 20 mM PBS and in culture media (diluted or suspended in PBS). In this case, when comparing the slopes we observed a decrease in the value of this parameter in the case of ML, respectively DLA. We assumed that the presence of aminoacides or other organic compounds has diminished the anodic signal, as can be previewed when working with real samples.
-page 10, Section 2.3 Evaluation of ENT signal in water samples: why the water samples were not analyzed using the in-lab printed electrodes on medical glove?
Answer: The electrochemical signal of standard solutions of Ent was lower in the case of in-lab printed electrodes and we attributed it to the fact that the electroactive surface was not submitted to any pretreatment after the printing protocol. In this case, to simplify and to achieve better recoveries, the commercial SPEs were used. Moreover, the use of in-lab printed electrodes was only a suggested application and further studies will be carried on when Ent will be assessed in biological samples, depending on the type of substrate selected for the application. The higher signal was obtained on the commercial SPE, and that is the reason that the water samples were tested on this type of electrode.
Taking into consideration the answer given by the authors, the water samples analysis using also the in-lab printed electrodes on medical glove would be even more significant, the comparison of the results being the basis of the proof of these electrodes applicability.
Answer: We acknowledge that the analysis of water samples with the in-lab printed electrodes would provide significant insight, however, based on the relatively low sensitivity of the electrode surface obtained for the in-lab printed electrodes in comparison with the commercial ones, we intend to further optimize the printing process and increase the sensitivity of the surface (by applying a preconditioning protocol or functionalizing with nanomaterials) and perform additional tests on water samples in future studies. At the moment, we are also limited by the access to more wastewater samples since they need to be collected and provided to us by the wastewater treatment plant. Moreover, the quality and properties of any kind of water samples would vary between different days and weeks, which would not reflect an accurate comparison with the data previously obtained (those included in this manuscript). The presented data serves as proof of concept for the applicability of the assessment of water samples with the in-lab printed electrodes, while the medical glove was designed for the analysis of clinical samples, which is foreseen for the near future and a future study.
- page 11, Figure 9: it would be more relevant if, instead of the obtained recoveries, Figure 9 would contain some voltammograms registered for the water samples, before and after spiking.
Answer: Figure 9 was modified according to the indication of the reviewer.
The differential pulse voltammograms named CW_blank and EW_blank are the voltammograms registered for the water samples before spiking and the other are those obtained after spiking with 25 μM, 50 μM and 75 μM Ent?
Answer: Yes, that is correct, the curves named with “_blank” were referring to the water samples before spiking. The legend and figure caption have been modified to be clearer.
In conclusion, my opinion is that the manuscript needs minor revision.